# Tilt Sensor with Recalibration Feature Based on MEMS Accelerometer

**DOI:** 10.3390/s22041504

**Published:** 2022-02-15

**Authors:** Sergiusz Łuczak, Maciej Zams, Bogdan Dąbrowski, Zbigniew Kusznierewicz

**Affiliations:** Warsaw University of Technology, Faculty of Mechatronics, 02-525 Warsaw, Poland; maciej.zams.dokt@pw.edu.pl (M.Z.); b.dabrowski86@gmail.com (B.D.); zbigniew.kusznierewicz@pw.edu.pl (Z.K.)

**Keywords:** tilt, MEMS accelerometer, recalibration, misalignment, compliant structure, drift, aging

## Abstract

The main errors of MEMS accelerometers are misalignments of their sensitivity axes, thermal and long-term drifts, imprecise factory calibration, and aging phenomena. In order to reduce these errors, a two-axial tilt sensor comprising a triaxial MEMS accelerometer, an aligning unit, and solid cubic housing was built. By means of the aligning unit it was possible to align the orientation of the accelerometer sensitive axes with respect to the housing with an accuracy of 0.03°. Owing to the housing, the sensor could be easily and quickly recalibrated, and thus errors such as thermal and long-term drifts as well as effects of aging were eliminated. Moreover, errors due to local and temporal variations of the gravitational acceleration can be compensated for. Procedures for calibrating and aligning the accelerometer are described. Values of thermal and long-term drifts of the tested sensor, resulting in tilt errors of even 0.4°, are presented. Application of the sensor for monitoring elevated loads is discussed.

## 1. Introduction

MEMS accelerometers are currently widely used in many electronic and mechatronic devices for a number of purposes primarily related to the sensing of linear acceleration, but also vibration [1], mechanical shock (jounce) [2], and one-axis tilt, i.e., inclination [3], or two-axial tilt [4]. Mathematical processing of the accelerometer output signal enables the sensing of linear velocity or displacement [5], such as in the case of applying MEMS accelerometers for a wheel odometer designed to determine the kinematics of a car [6]. While employing a higher number of accelerometers with a strictly defined geometrical configuration, accelerometers can replace gyroscopes in the sensing of angular rates [7]. 

Despite numerous advantages of these microsensors, including low price, robustness, small size, high shock-resistance and easy integration with electronics, there are a few disadvantages that characterize their performance. These are mainly: misalignments of the sensitivity axes (including their non-perpendicularity) [8]; thermal and long-term drifts of the output signals (affecting both the offset and the scale factor) [9]; errors related to factory calibration of the accelerometer and aging of the silicon structure [8]; and attenuation of amplitude and phase shift over frequency [10].

Misalignments of the accelerometer sensitivity axes are very harmful [11,12], especially in the case of performing accurate tilt measurements [13], which are one of the most typical applications of MEMS accelerometers [14]. Therefore, many solutions have been proposed to address this problem, mainly based on numerical compensation for the existing misalignment angles, as proposed, e.g., in [15,16,17].

Thermal drifts of the offset and the scale factor, of which the first is much more dominant [9], are dealt with by the application of temperature sensors and using averaged thermal characteristics (or look-up tables) of particular accelerometer models. Another approach is keeping the accelerometer at a constant ambient temperature [9], e.g., by means of the application of micro-ovens [18], either custom or standard—as in the case of precise quartz-crystal resonators [19].

Long-term drifts and effects of aging of the silicon structure are more difficult to be compensated for. Some manufacturers declare maximal changes due to aging over the whole lifetime of a MEMS device [20], and other manufacturers implement models for the prediction of these changes [8]; overall though, the long-term drifts are rather difficult to predict. It seems that repetition of the calibration process is the only reasonable solution which enables a significant reduction of the related errors.

Moreover, recalibration of the sensor may compensate for errors resulting from such factors as geographic latitude, altitude, and local and temporal changes of gravitational acceleration due to many other reasons (e.g., constellation of the heavenly bodies, especially the sun and moon).

One of the ways to eliminate or compensate for such errors as unknown orientation of the accelerometer sensitivity axes, thermal and long-term drifts of the output signals, calibration errors of the accelerometer, and aging of the silicon structure is sensor recalibration with the use of a test stand [21,22,23], or by performing auto-calibration [24,25], allowing the sensor operating parameters to be accurately determined under given operating conditions. Using a test stand is of course expensive and laborious, whereas auto-calibration is usually complicated, consumes a lot of computing power, and may require a sophisticated analysis of the existing errors, e.g., in [26].

It was decided to evaluate how effective, in terms of the elimination of some of the relevant errors, a recalibration of MEMS accelerometers may be, assuming that no dedicated test stand would be used. It was proposed to reduce the test stand to a flat leveled surface and build a special tilt sensor based on a triaxial MEMS accelerometer, since this enables performance of dual-axis tilt measurements within full measurement range, i.e., 2 × 360° [27]. The tilt sensor is equipped with a custom housing that makes it possible to repeat a simple and quick calibration procedure of the accelerometer, and comprises an aligning unit for the reduction of the misalignments of the accelerometer sensitive axes. 

The experimental results proved that the proposed tilt sensor, characterized by relatively low cost, features accuracy much higher than the results from the parameters of the applied MEMS accelerometer. Misalignment of the sensitive axes did not exceed 0.03° and, owing to recalibration, thermal and long-term drifts of the output signals were eliminated. At the same time, the following errors were fully eliminated: imprecise factory-calibration of the offset and the scale factor for each sensitivity axis of the accelerometer, as well as aging effects of the sensor-chip.

## 2. Mechanical Structure of the Device

As aforementioned, the essential shortcomings of MEMS accelerometers can be eliminated by repeating their calibration process. When implementing a simple calibration procedure, which does not require the application of a dedicated test stand, it is beneficial to use suitable accelerometer housing (which typically has an unfavorable geometry), preferably of a cubic shape. Such an idea has been presented in [28], where it was proposed to equip the tested inertial measurement unit (IMU) with a rectangular housing adapted to calibrate the component triaxial MEMS accelerometer and MEMS gyroscope. On the other hand, in [13] a special mechanical unit used for alignment of the used MEMS accelerometer is described. In this paper, a sensor using both solutions is presented, where the aligning unit, used to adjust orientation of the MEMS accelerometer, is mounted in a housing that allows its calibration to be performed without a test stand. 

### 2.1. The Housing

In order to replace the role of the test stand during the calibration of the sensor, the housing must have a specific geometrical shape. Maintaining, at the same time, a good alignment of the accelerometer sensitive axes with respect to the external mechanical frame (housing) significantly reduces the influence of the related errors. The presented mechanical structure was designed to ensure possibly simple realization of these two ideas. Therefore, the housing of the sensor has a cubic shape, as presented in Figure 1, with a chamber open in one of the faces, in which the chamber an aligning unit is installed. The aligning unit supports a printed circuit board, onto which a MEMS accelerometer is soldered. 

The housing is made of stainless steel, which ensures sufficient and durable perpendicularity of its faces, with the respective deviation of ca. 0.01 degrees arc, which results from precession of its machining. It is possible to obtain lower deviation of perpendicularity; however, this is at considerably higher machining costs, especially in the case of stainless steel. Nevertheless, if the housing is expected to be used for determining the inherent misalignments of the accelerometer sensitive axes (e.g., when the axes are not perpendicular, as discussed later in the text), deviation of perpendicularity of its faces must not exceed the expected alignment error.

The housing has three openings created in its faces. The first opening provides a space for the aligning unit and the accelerometer PCB. The structure of the housing facilitates placement of the unit.

The second opening of a cylindrical shape in the housing is created for a spirit level, which makes it easier to level the flat surface used during calibration of the sensor, proving a leveling precision of ca. 1 min arc.

The third opening is a small notch, which enables the cable to be put on different faces, depending on the housing orientation during calibration of the sensor. The cable transmits measurement data from the MEMS accelerometer and supplies it with power. Alternatively, the sensor may be equipped with a wireless communication module connected with the accelerometer and adapted to transmit its readings to an external computing unit, just as proposed in [28] and presented in Section 7. 

Additionally, several groups of threaded holes are provided in the housing. The first allows the sensor to be mounted in the end-user device or to be fixed in a special test stand while aligning the accelerometer sensitive axes. The second makes it possible to secure all the sensor components in the housing (the aligning unit, the spirit level, a grip of the cable), whereas the third is for the aligning screws. 

The constructed tilt sensor is based on using a triaxial MEMS accelerometer; therefore, it can be used for two-axial tilt sensing. Of course, it is also possible to sense single-axis tilt (i.e., inclination: either pitch or roll) or axial tilt. Then, the triaxial MEMS accelerometer may be replaced by a biaxial or uniaxial one, as discussed in [27].

In order to determine perpendicularity deviation of the adjacent faces of the machined housing, a precise coordinate measuring machine (CMM) was employed. The maximal deviation of perpendicularity *DP* among all the adjacent faces of the housing was 36 s arc (0.01°), whereas deviation of flatness did not exceed 5 μm.

However, it should be noted that for alignment purposes, it is sufficient to make the geometrical data of only two adjacent faces of the housing. The faces should be flat and approximately perpendicular: the first is the base that removes two rotational degrees of freedom of the sensor, and the second removes the third rotational degree of freedom. 

Nonetheless, if all the faces of the housing are machined precisely, any pair of adjacent faces can be used as the geometrical data, making the installation of the housing in the end-user device more convenient.

### 2.2. The Aligning Unit

The aligning unit, presented in Figure 2 in top view, is fixed to the housing by two clamping screws, and its orientation with respect to the housing is determined by three aligning screws oriented approximately perpendicularly to each other. Between the mounting section of the aligning unit and its supporting section, a U-shaped groove is created partially surrounding the mounting section. The PCB with MEMS accelerometer is fixed to the supporting section by means of four clamping screws. Owing to the groove, it is possible to elastically deform the aligning unit in such a manner that the supporting section can reversibly move with respect to the mounting section. The angular displacements can be controlled by means of three aligning screws seated in the walls of the housing. Thus, it is possible to align the accelerometer by changing the orientation of the supporting section of the unit with respect to the housing.

The three aligning screws are seated in threaded holes created in respective walls of the housing. The aligning screw *5* rests against the supporting section which, owing to the U-shaped groove, may be deflected with respect to the mounting section. Screwing in this screw results in rotation of the supporting section about *y*-axis. The aligning screw *4* is parallel to the aligning screw *5*. It is located within the same plane of the unit, but is closer to the mounting section. Screwing in this screw results in rotation of the supporting section about *x*-axis. The third aligning screw *3* is oriented perpendicularly to the other two. Screwing in this screw results in rotation of the supporting section about *z*-axis. Thus, the presented configuration provides adjustment of accelerometer orientation around three perpendicular axes. 

The aligning unit is made of aluminum alloy featuring good elasticity and high fatigue strength. Nevertheless, the unit may be made of some other material, preferably (due to good elastic properties): beryllium bronze, brass, steel, polymer (including 3D printing), composites (glass or carbon fiber).

In order to reveal a special relief created next to the groove, the aligning unit is shown in bottom view in Figure 3. 

The relief was introduced in order to reduce the mechanical stress in the structure of the aligning unit. As can be seen in Figure 4, the maximal value of the Von Mises stress, while fully loaded by means of all three aligning screws, does not exceed 155 MPa, which is a relatively high, yet acceptable with respect to aluminum alloy.

It must be realized that it is almost impossible to build such an aligning mechanism that would ensure no interaction between alignments in particular axes—especially when keeping its overall dimensions small. Therefore, the alignment procedure, being of experimental nature, must be iterative, i.e., additional fine alignment in one axis must follow a rough alignment in the other axes. Typically, 2 or 3 iterations ensure a satisfactory result.

## 3. Physical Alignment of the MEMS Accelerometer

Physical alignment enables a precise positioning of the accelerometer with respect to the sensor housing, i.e., the particular accelerometer sensitivity axis is oriented in such a way that it is either parallel or perpendicular to the housing external faces.

While aligning the accelerometer, a simple procedure, minutely described in [13], can be used. It basically consists of slowly rotating the sensor about a rotation axis parallel to the respective housing face (which is also approximately parallel to the sensitive axis being aligned), and monitoring the output signal associated with the sensitive axis being aligned. Variations of the output signal should be minimized by changing the orientation of the sensitive axis by means of two appropriate aligning screws.

First, the tilt sensor must be precisely fixed in a test stand, ensuring parallel orientation of the respective housing face with respect to the rotation axis of the test stand. Then, the aligning procedure should be performed for the *X*-axis (see Figure 2), setting aligning screws *3* and *5* to obtain minimal variations of the *x*-output signal. Then, the aligning procedure should be performed for the *Y*-axis. However, at this time only the aligning screw *4* can be used in order to obtain minimal variations of the *y*-output signal. Alignment precision of the *Z*-axis results from the two procedures. The order of aligning of the sensitive axes can be of course changed, according to particular demands related, e.g., to the accuracy of tilt measurements. 

It must be realized that a complete elimination of all misalignments is generally impossible due to the fact that accelerometer sensitive axes are not perfectly perpendicular. The inherent mutual misalignments may be much higher than specified in the relevant datasheets, even of 1° [27]. This problem is discussed in Section 6.

### The Alignment Precision

In order to precisely align the accelerometer with respect to the housing, the test rig presented in Figure 5 was employed. The housing of the aligned sensor was secured in a special precise fixture installed on the output shaft of the optical rotary head. The shaft was driven manually with the positioning accuracy of a 3 s arc. The accelerometer was supplied by a stabilized power unit and constant-voltage regulator (nominal voltage: 3 V; range of variation of the voltage throughout the whole experiment did not exceed 0.25 mV), and its output analog voltage signal was sampled by means of a data acquisition module USB 6211 by National Instruments (not shown in Figure 5) and recorded in computer memory.

According to the method described in [13], a series of 30 measurements were performed at the following angular positions, the resultant average value was observed, and the aligning unit was adjusted accordingly. After few revolutions, it was possible to align the particular sensitive axis so precisely that the observed variation of the averaged output voltage did not exceed ±*LSB* [13], where *LSB* is the least significant bit (in the case of the 16-bit USB 6211 device and the measurement range of 10 V: *LSB* = 0.15 mV). Thus, the error of the alignment procedure *APE* can be calculated as follows:(1)$$APE=\arcsin\left(\frac{LSB}{SF}\right)=\arcsin\left(\frac{0.00015}{0.41}\right)=0.021^{{}^{\circ}},$$
where *SF* is the scale factor of the accelerometer (see Section 5.1). 

On the other hand, the total alignment error (*TAE*) can be evaluated as follows:(2)$$TAE=APE+DP\approx 0.02^{{}^{\circ}}+0.01^{{}^{\circ}}\approx 0.03^{{}^{\circ}},$$
where *DP* is the deviation of perpendicularity (see Section 2.1).

## 4. Calibration of the Sensor

As aforementioned, owing to the cubic shape of the housing, a test stand was not necessary to perform the calibration. The only element required was a platform that could be precisely leveled, e.g., using the embedded spirit level. 

Various concepts of the low-g accelerometer calibration process have been proposed in relevant publications, e.g., using twelve characteristic orientations of the accelerometer, employing a simple instrument such as a three-way milling vice [21]. However, in most of the related research, only six characteristic orientations were employed, e.g., in [9,23,28]. The accepted idea of the calibration was also based on the latter concept, and was very simple. It was based on recording the sensor output signals under static conditions at six specific orientations, where one of the accelerometer output signals reached its extreme (maximal or minimal) value. The six positions were, of course, defined by the faces of the housing and are presented in Figure 6.

In order to perform the proposed type of calibration, the accelerometer output signals were measured after it was placed on each of the six external faces of the housing. At each position, a series of measurements of the output signal associated with a particular axis was performed. Then, an average value was determined for each axis at each position.

In the case of the reported study, 300 measurements of the analog voltages were performed per each axis, at each of the six calibration positions. For this purpose, the data acquisition module USB 6211 by National Instruments was used, whose absolute accuracy at the selected scale range (10 V) was specified as 2.7 mV [29] (it should be noted that the relevant accuracy was higher, since the performed measurements reached only 20% of the measurement scale and no variation of the ambient temperature was involved). For all of the 18-measurement series, the type-A uncertainty [30] did not exceed 0.24 mV. Assuming Gaussian distribution, the expanded uncertainty (3-*σ* error), corresponding to a probability of 99.73%, was equal to 0.72 mV. 

At this point, it is worthwhile to mention that at the orientations illustrated in Figure 6, the calibrated accelerometer axis is not sensitive to relatively large inclinations of the housing. Let us assume that the reference surface is precisely leveled (accuracy of 1 min arc), and the housing is characterized by a large theoretical deviation of perpendicularity *TDP* as high as 1 degree arc. As a result, the calibrated axis, instead of the gravitational acceleration *g*, senses its component that is equal to *g*cos(*TDP*), which is only 0.02% smaller than *g.* The resultant calibration error (*CE*) expressed as tilt angle can be evaluated as follows:(3)$$CE=\arcsin\left[1-\cos\left(TDP\right)]=\arcsin[1-\cos\left(1^{{}^{\circ}}\right)\right]=0.0087^{{}^{\circ}},$$
where *TDP* is a theoretical deviation of perpendicularity, accepted as 1°.

So, from the point of view of the calibration, deviation of perpendicularity of the housing being as large as 1 degree arc results in a tilt error below a 0.01 degree arc. So, if only two faces of the housing are to be used for aligning the sensor with respect to the frame of the end-user device, as suggested in Section 2.1, deviation of perpendicularity of all the faces of the housing may be even that large. 

The purpose of the calibration process is to determine two individual parameters for each sensitive axis: offset (bias) *OF* and scale factor *SF*. Having all the data recorded, maximal and minimal average value for each sensitive axis must be found. Then, the following equations can be used to compute the set of the offsets and scale factors:(4)$$OF_{x}=\frac{\max\left(U_{x}\right)+\min\left(U_{x}\right)}{2},$$
(5)$$SF_{x}=\frac{\max\left(U_{x}\right)-\min\left(U_{x}\right)}{2},$$
(6)$$OF_{y}=\frac{\max\left(U_{y}\right)+\min\left(U_{y}\right)}{2},$$
(7)$$SF_{y}=\frac{\max\left(U_{y}\right)-\min\left(U_{y}\right)}{2},$$
(8)$$OF_{z}=\frac{\max\left(U_{z}\right)+\min\left(U_{z}\right)}{2},$$
(9)$$SF_{z}=\frac{\max\left(U_{z}\right)-\min\left(U_{z}\right)}{2},$$
where *U_x_.._z_* are averaged voltage output signals assigned to particular sensitive axis of the MEMS accelerometer, expressed in [V]; *OF_x_.._z_* are offsets (biases) associated with particular sensitive axis, expressed in [V]; and *SF_x_.._z_* are scale factors associated with particular sensitive axis, expressed in [V/*g*], where *g* is the gravitational acceleration.

Once the calibration parameters have been determined, accelerations indicated by the accelerometer can be expressed as a fraction of *g* (referring to tilt measurements), using the following formulas: (10)$$a_{x}=\frac{U_{x}-OF_{x}}{SF_{x}}\;\left[g\right],$$
(11)$$a_{y}=\frac{U_{y}-OF_{y}}{SF_{y}}\;\left[g\right],$$
(12)$$a_{z}=\frac{U_{z}-OF_{z}}{SF_{z}}\;\left[g\right].$$

In order to calculate tilt (expressed as pitch, roll or axial tilt), various formulas based on inverse trigonometric functions can be used, as discussed in [27]. 

It should be noted that owing to the recalibration of an accelerometer at the location of its operation, the errors due to variations of the gravitational acceleration (resulting mainly from the local altitude and latitude) can be eliminated. If the recalibration is cyclically repeated, even temporal variations of the gravitational acceleration (e.g., resulting from interactions of the heavenly bodies) can be considerably reduced.

### 4.1. Misalignment Angles

If it is impossible to precisely align the accelerometer (e.g., due to its inherent imperfections), the component misalignment angles can be determined (two for each axis) during the calibration. In such case, not only the extreme values of the output signals of the accelerometer must be used, but all the average values. At a position of the sensor when a particular sensitive axis has approximately horizontal orientation, the associated output signal makes it possible to determine the respective misalignment angle, which the accelerometer senses as pitch or roll. 

Referring to Figure 2 and Figure 6, the respective positions will be the following for the particular sensitive axes: pos. 1 and pos. 2 and pos. 3 and pos. 4 for *x*-axis;pos. 3 and pos. 4 and pos. 5 and pos. 6 for *y*-axis;pos. 1 and pos. 2 and pos. 5 and pos. 6 for *z*-axis.

Using the scales factors and offsets determined in the first step, it is possible to determine the misalignment angles *MA* using the following formulas:(13)$$MA_{1x}=\arcsin\left(\frac{U_{1x}-U_{2x}}{2SF_{x}}\right),$$
(14)$$MA_{2x}=\arcsin\left(\frac{U_{3x}-U_{4x}}{2SF_{x}}\right),$$
(15)$$MA_{1y}=\arcsin\left(\frac{U_{3y}-U_{4y}}{2SF_{y}}\right),$$
(16)$$MA_{2y}=\arcsin\left(\frac{U_{5y}-U_{6y}}{2SF_{y}}\right),$$
(17)$$MA_{1z}=\arcsin\left(\frac{U_{1z}-U_{2z}}{2SF_{z}}\right),$$
(18)$$MA_{2z}=\arcsin\left(\frac{U_{5z}-U_{6z}}{2SF_{z}}\right),$$
where *U*_1_.._6*x*_.._*z*_ are averaged voltage signals assigned to a particular sensitive axis (*x*..*z*) at a given position (1..6 as in Figure 6), expressed in Volts.

## 5. Thermal and Long-Term Drifts

To evaluate the performance of the constructed tilt sensor, a triaxial MEMS accelerometer ADXL 327 [31] was employed. Once the accelerometer had been aligned, the sensor could have been calibrated.

The following methodology was accepted while studying thermal and long-term drifts of the accelerometer. The full calibration procedure, as described in Section 4, was repeated cyclically. Each time, a series of the accelerometer output signals was measured after placing it on each of the six faces of the housing. Then, using averaged values of the measurements, offset and scale factors for each sensitive axis were determined. The ambient temperature was kept constant both while determining the thermal drifts as well as the long-term drifts. The difference was the time interval between the successive recalibrations. In the first case it was 1, 2, 4 and 6 h., whereas in the second case it was 24 and 48 h.

### 5.1. Thermal Drifts

The first experiment was performed in order to reveal thermal drifts resulting from self-heating of the accelerometer, just after its start-up. The ambient temperature was kept constant. A relevant publication reports on the considerable temperature changes after the start-up of the accelerometer (up to 3 °C) [9]. The changes are characterized by continuous temperature increase over the 5 h. of testing.

So, the sensor was supplied with power, and the first calibration was performed. Then, calibration was repeated after 1, 2, 4 and 6 h. Results of the measurements of output signals associated with particular sensitive axes against the time since the sensor was powered on are presented in Table 1, Table 2, Table 3, Table 4 and Table 5. The ambient temperature was kept constant during the experiment.

Results of the calculations of the offset *OF* and scale factor *SF* for particular sensitive axis against the time since the sensor was powered on, are presented in Table 6 and Table 7. The last row indicates the maximal absolute value of the variations of each parameter within 6 h.

As can be observed, both at the reheating phase and during early operation of the device, just after connecting it to the power supply, there were no significant temperature fluctuations, since the sensor parameters were not considerably affected. The observed changes were not systematic. Once the parameters increased, and once decreased: no continuous trend can be stated. This observation is altogether different than in the case of a similar study reported in [9]; this can be explained by the application of different types of accelerometers in both studies (in the case of the cited study, the tested accelerometer was installed on a PCB together with a microprocessor unit generating additional heat). Another issue was a good heat transfer from the accelerometer, since the aligning unit is made of aluminum and has relatively large surfaces, besides, the accelerometer is also exposed to open air. 

It was not planned to study the thermal drifts of the accelerometer resulting from changes of the ambient temperature, since values of related parameters are provided by the manufacturer in the dedicated datasheet [31]. Nevertheless, it is worthwhile mentioning that the drift of the offset was more significant than the drift of the scale factor [9].

### 5.2. Long-Term Drifts

The same procedure used for determining the thermal drifts was adapted for studying long-term drifts. The calibration process was performed at the beginning of the test, then it was repeated after 24 and 48 h. of continuous operation of the sensor under constant ambient temperature. Variations of the offset and the scale factor for particular sensitive axes are illustrated in Figure 7.

Table 8 presents relative values of the variations within 48 h. of each offset and scale factor illustrated in Figure 7. The last row indicates the maximal absolute values. 

Analyzing the results illustrated both in Figure 7 and Table 8, it can be observed that the variations are rather of a random character. Each sensitive axis reveals its own individual trends. The highest variation of 0.2% was observed for *z*-axis, since this is rather predictable, due to the fact that this axis generally features worse metrological parameters [27]. The main reason for such a situation is the fact that the MEMS manufacturing processes are in fact only semi-three-dimensional [20], often resulting in a worse performance in the vertical axis. Of course, this is not a general rule, and there exist designs of triaxial (or even multiaxial) MEMS accelerometers featuring uniform parameters in all sensitive axes, taking for an example the original design presented in [32].

### 5.3. Resultant Errors 

In order to evaluate the value of the maximal relative error of acceleration measurement, which results from the observed maximal long-term drift, affecting both the offset and the sale factor, the following formula should be applied [8]:(19)$${\Delta g}_{zl}=\;\Delta SF_{z}+\frac{\Delta OF_{z}\cdot OF_{z}}{SF_{z}}=0.74\%,$$
where Δ*g_zl_* is the relative error (with respect to *g*) of acceleration measurement in *z*-axis due to long-term drift. 

In analogy to Equation (1), the value of the resultant maximal tilt error *TEL* can be calculated as follows:(20)$$TEL=\arcsin\left({\Delta g}_{zl}\right)\approx 0.43^{{}^{\circ}}.$$

Similar procedure can be applied to errors resulting from aging *TEA*. Even though aging effects were not determined in this study, we refer to experimental results related to the same accelerometer ADXL 327 tested over 4.5 years, presented in [8]. Analogously to Equation (20):(21)$$TEA=\arcsin\left({\Delta g}_{za}\right)\approx 0.86^{{}^{\circ}}.$$
where Δ*g_za_* is the maximal relative error (with respect to *g*) of acceleration measurement in the *z*-axis due to aging phenomena, equal to 1.5% [8]. 

The above values of errors are considerable as related to tilt errors, especially in view of the fact that the long-term drifts were determined over only a 48-h period, and the errors due to aging over only 4.5 years–longer periods may be the reality. 

## 6. Discussion

Since MEMS accelerometers may be characterized by the considerable deviation of perpendicularity of their sensitive axes, the presented mechanical structure may not be sufficient in some cases, since it allows only one of the sensitive axes to be fully aligned with respect to the housing of the sensor. Consequently, the second sensitivity axis can be precisely aligned only within one plane, and the third cannot be aligned without losing the alignment of the previous two. In such cases, the following solutions are possible:Application of a more precise MEMS accelerometer, with low deviation of perpendicularity of its sensitive axes (either different model or a better piece from the same production lot);Using only two appropriate sensitive axes (at the cost of decreasing the measurement range or/and the sensitivity, as discussed in [27]);Using two or three MEMS accelerometers—each having its own aligning unit, featuring only two degrees of freedom (i.e., employing only two aligning screws). In other words, developing a more complicated mechanical structure of the sensor;Having the first sensitive axis fully aligned, the second partially aligned, and then determining in an experimental way during recalibration the remaining component misalignment angles: one for the second and two for the third sensitive axis, as described in Section 4.1;Not using the aligning unit and applying instead numerical models compensating for the existing misalignments of all the sensitive axes (i.e., three angles related to non-perpendicularity of each sensitive axis—as proposed in [16]—or six component misalignment angles, two for each sensitive axis—as proposed in [15]).

Each solution had its own advantages and disadvantages. The most interesting one seems to be the mechanical structure with three MEMS accelerometers and three aligning units, where it is possible to eliminate all the misalignments (i.e., two component angular displacements per each sensitive axis). 

## 7. 3D-Printed Housing

Because of a high cost of machining for the housing made of stainless steel, it was decided to use additive manufacturing instead. Two prototypes were made of acrylonitrile butadiene styrene (ABS), using fused material deposition (FDM) technology. One of the manufactured prototypes of the tilt sensor is presented in Figure 8. Moreover, it was decided to add a module of wireless transmission and a battery in order to eliminate the cable. 

The prototype had dimensions of 40 × 48 × 48 mm [33], the same as in the case of the steel housing. However, it was not equipped with a spirit level, and employed a different design of the aligning unit: instead of a compliant structure, a commercial cap-and-ball joint was used. Nevertheless, the alignment principle remained the same—it was realized by means of 3 screws and small helical springs as their counterparts.

Because of a considerable shape deviation of the printed housing, its footing had to be machined in order to make it flat, which is important while calibrating the tilt sensor. While using additive manufacturing, attention must be paid to the selection of an appropriate material featuring possibly high mechanical strength in the first place. Then, an appropriate 3D printing technique must be used. As proven by experimental tests of 3D prints, mechanical properties of various materials differ considerably [34,35]. 

In the case of the second prototype, an attempt was made to manufacture the ball of the joint as integrated with the housing [36]. However, due to porous structure of the 3D prints, it broke easily. Moreover, the ball and its counterparts were characterized by higher friction compared to the commercial cap-and-ball joint. So, because of material properties, it is recommended to use additive manufacturing only for the housing, and to machine the aligning unit in a conventional way. Nonetheless, another trade-off must be realized: substituting stainless steel with a printed polymer results also in a significant worsening of the robustness of the housing. This may not be acceptable when the tilt sensor is operated under harsh industrial conditions. Still, another problem is the limited temperature range, both for the operation of as well as the storing of 3D prints. Our own tests proved that even the temperature inside a car on a hot summer day (ca. 60 °C) causes a permanent distortion of 3D prints if exposed for a long time. These prints are also hygroscopic, and it is difficult to define the long-term stability of their shape and dimensions.

Nevertheless, when the tilt sensor is to be used only under laboratory conditions (research, didactic classes), printed housing may be considered as a much cheaper option compared to steel housing. Additional machining of the footing ensures satisfactory flatness, whereas linear accuracy of FDM of ca. 0.1 mm is high enough with respect to the required deviation of perpendicularity of the housing faces, which under special conditions may be as high as 1° (see explanation in Section 2.1). As in the case of the steel housing, perpendicularity deviation of the adjacent faces of the two ABS housings was determined by the coordinate measuring machine. The maximal deviation of perpendicularity among all the adjacent faces of the housing was approximately of few min arc (0.1°).

## 8. Monitoring of Tilt of Elevated Loads

In order to maintain the high alignment accuracy of the accelerometer sensitive axes while employing the tilt sensor in the monitoring of elevated loads, a special mechanical device was designed. Basically, this provides the housing of the tilt sensor with two angular degrees of freedom, which are controlled by means of two aligning screws, seated in threaded holes of the two-part housing, using the same principle of operation as in the case of the unit aligning the MEMS accelerometer. By setting the aligning screws, it is possible to align appropriate planes of the housing of the tilt sensor with the axis of the lifting rope, as illustrated in Figure 9. 

Because of the fact that the direction of the alignment rotation is random, the respective walls should not be perpendicular, but the angles alpha and beta, illustrated in Figure 10, should be slightly bigger than the right angle (e.g., of 92 degrees arc).

While determining the maximal axial tilt of the elevated load that constantly swings on the lifting rope, the following inverse trigonometric functions can be used:(22)$$\varphi_{max}=\arccos\left(\frac{\min\left(a_{z}\right)}{g}\right),$$
(23)$$\varphi_{max}=\arcsin\left(\frac{\sqrt{\max{\left(a_{x}\right)}^{2}+\;\max{\left(a_{y}\right)}^{2}}}{g}\right),$$
(24)$$\varphi_{max}=\arctan\left(\frac{\sqrt{\max{\left(a_{x}\right)}^{2}+\;\max{\left(a_{y}\right)}^{2}}}{\min\left(a_{z}\right)}\right),$$
(25)$$\varphi_{max}=\arccos\left(1.5-\frac{\max\left(a_{z}\right)}{2g}\right),$$
where *a_x_*, *a_y_*, *a_z_* are the Cartesian components of the acceleration sensed by the applied MEMS accelerometer, and *g* is the gravitational acceleration.

## 9. Summary

An original tilt sensor has been presented. Its parameters are:Measurement range: 360-degree arc about two axes (pitch and roll angle measurement over the full range);Misalignments of the sensitive axes with respect to the housing of approx. 0.03 degree arc;Output signals in the form of analog voltages of approx. 1–3 V or digital (depending on the applied MEMS accelerometer or its operation mode);Dimensions of the PCB with MEMS accelerometer: 22 × 27 × 3 mm;Housing dimensions: 40 × 48 × 48 mm (volume: 93 cubic cm);Housing material: stainless steel (ensuring robustness) or 3D-printed polymer (ensuring low cost);Mechanical datum of the housing for accurate mounting of the sensor onto the end-user device, using several threaded holes created in the housing;External faces of the housing allowing accurate calibration and alignment of the sensor to be performed on a leveled surface;Error of leveling the housing (by means of the embedded spirit level) of approx. 1 min arc.

The presented tilt sensor, owing to the special design of its housing, can be easily and quickly recalibrated by the user in order to eliminate such considerable errors as thermal and long-term drifts as well as effects of aging, keeping the sensitive axes aligned with respect to the external mechanical datum (two faces of the housing).

As it has been experimentally discovered, thermal drifts due to self-heating of the accelerometer typically resulted in relatively insignificant variations of the offset or the scale factor of approx. 0.02%. However, long-term drifts resulted in variations of these parameters as high as 0.2% (3 mV) within only 48 h., corresponding to the acceleration error of ca. 0.7% and tilt error of ca. 0.4°.

In order to compare the performance of the proposed sensor, misalignment values of four inertial measurement units (IMU) with 6–10 degrees of freedom (DOF) and a MEMS accelerometer by Analog Devices Inc. are listed in Table 9. It should be noted that it is difficult to find MEMS sensors with misalignments of the sensitive axes with respect to the frame (packaging) specified separately in the respective datasheet—this is why sensors by Analog Devices Inc. were selected.

Moreover, the manufacturer stated that they strove for a tight orthogonal alignment of the sensitive axes, since it simplifies the alignment of the sensor packaging in navigation systems [37,38]. The listed mutual misalignment of the sensitive axes is an inherent feature of a given sensor, and cannot be physically reduced. 

While comparing the values of misalignments of the sensitive axes with respect to the packaging (minimal value of ±0.5°), it can be stated that the corresponding value featured by the proposed sensor (approximately ±0.03°) was considerably lower (16 times).

The IMUs listed in Table 9 had their thermal drifts compensated for. However, even though some of them were quite expensive (price over USD 1000), the long-term drifts and effects of aging phenomena were not reduced, whereas the proposed sensor makes it possible to fully eliminate these effects owing to the recalibration process.

## 10. Patents

As a result of the work related to the discussed content, a patent [39] was submitted to the Polish Patent Office. It described both the aligning unit as well as the special housing enabling the recalibration function. Another related patent [40] was submitted to the Polish Patent Office later. It addressed a specific application of the proposed sensor for monitoring the dynamic tilt of elevated loads.

## Figures and Tables

**Figure 1 sensors-22-01504-f001:**
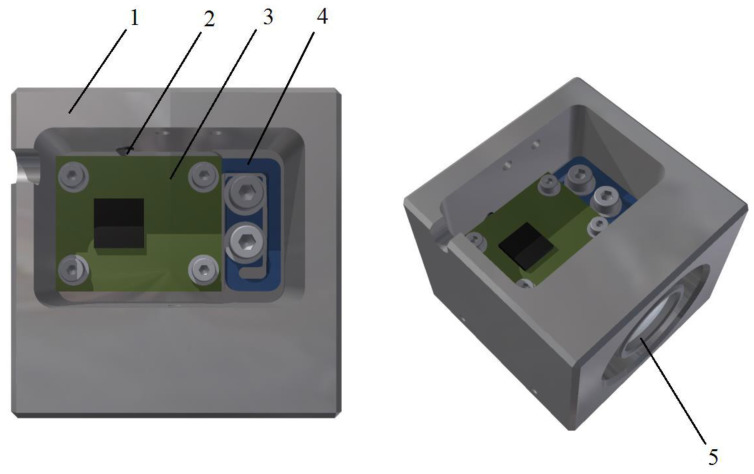
Structure of the tilt sensor: 1—housing; 2—aligning screw, 3—PCB with MEMS accelerometer; 4—aligning unit; 5—spirit level.

**Figure 2 sensors-22-01504-f002:**
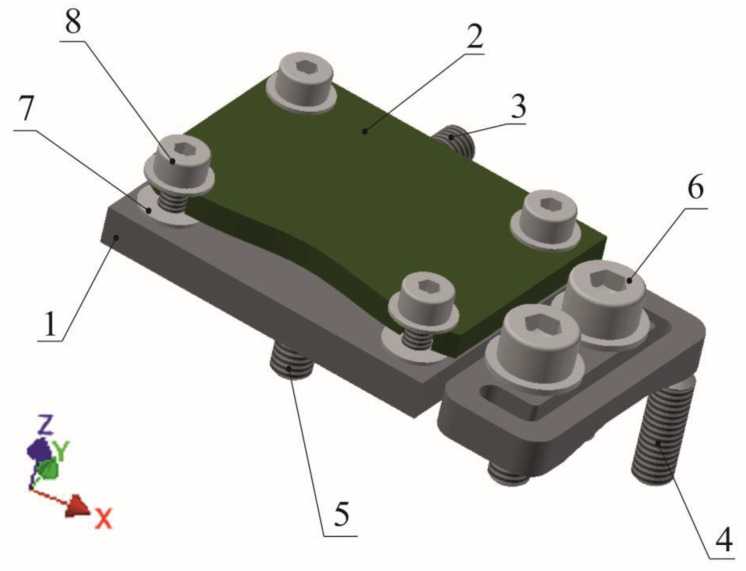
Structure of the aligning unit (top view): 1—supporting section; 2—PCB with MEMS accelerometer; 3, 4, 5—aligning screws; 6, 8—clamping screws; 7—leveling washers.

**Figure 3 sensors-22-01504-f003:**
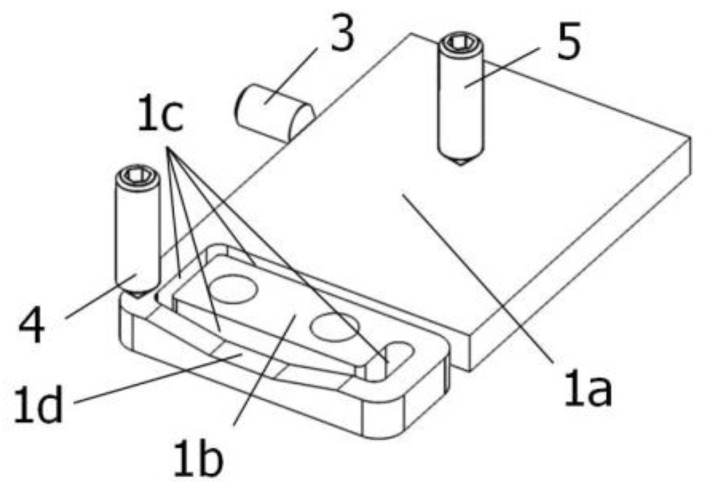
Bottom view of the aligning unit: 1a—supporting section; 1b—mounting section; 1c—U-shaped grove; 1d—relief; 3, 4, 5—aligning screws.

**Figure 4 sensors-22-01504-f004:**
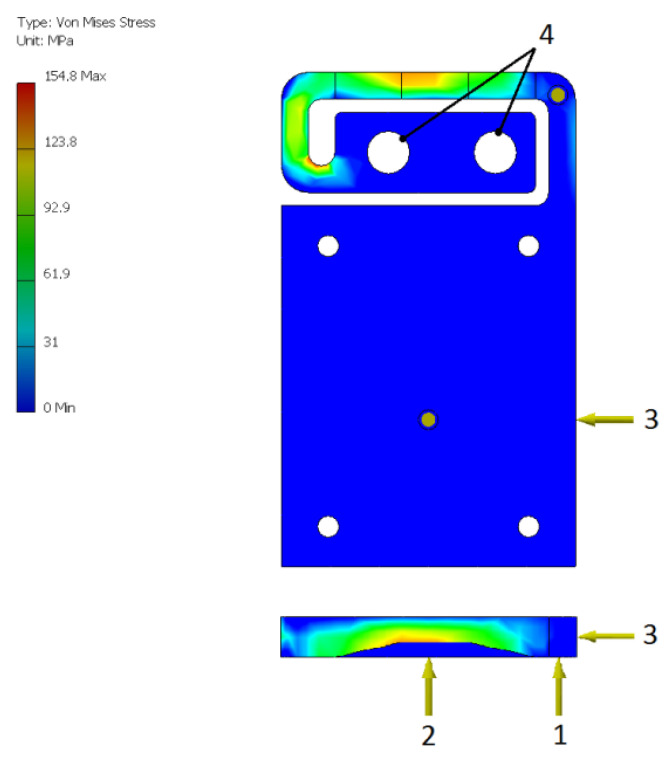
Mechanical stress within the aligning unit.

**Figure 5 sensors-22-01504-f005:**
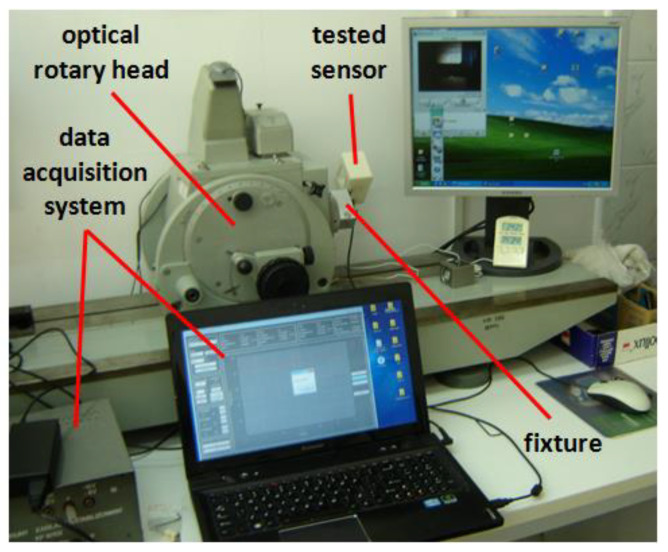
Experimental test rig during the alignment procedure.

**Figure 6 sensors-22-01504-f006:**
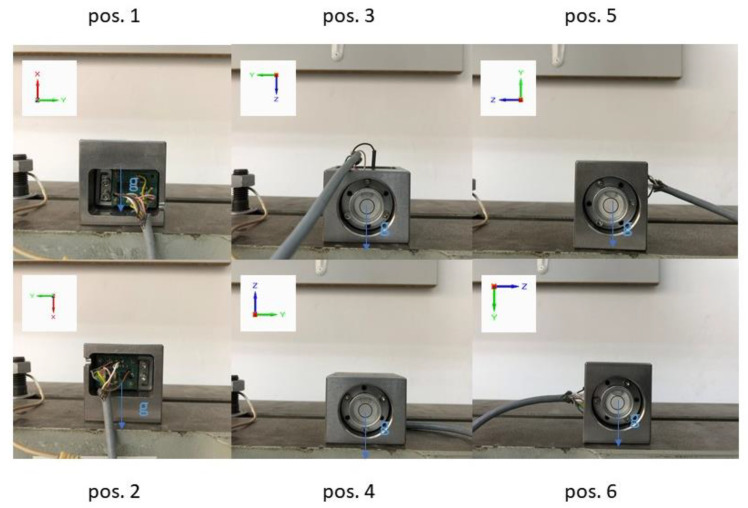
Six orientations of the sensor during calibration.

**Figure 7 sensors-22-01504-f007:**
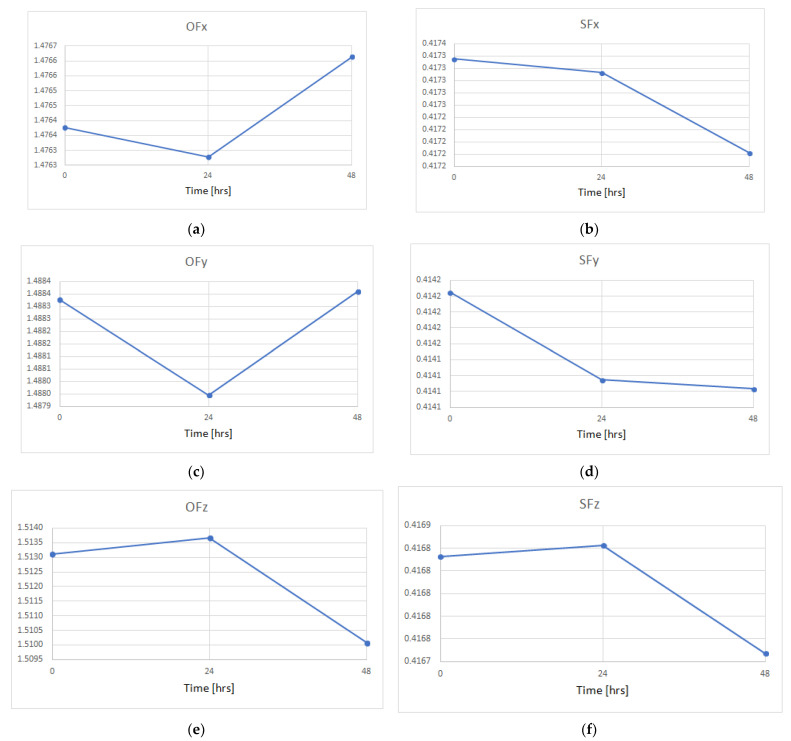
Long-term drifts of the offset and the scale factor over 48 h: (**a**) offset of *x*-axis; (**b**) scale factor of *x*-axis; (**c**) offset of *y*-axis; (**d**) scale factor of *y*-axis; (**e**) offset of *z*-axis; (**f**) scale factor of *z*-axis.

**Figure 8 sensors-22-01504-f008:**
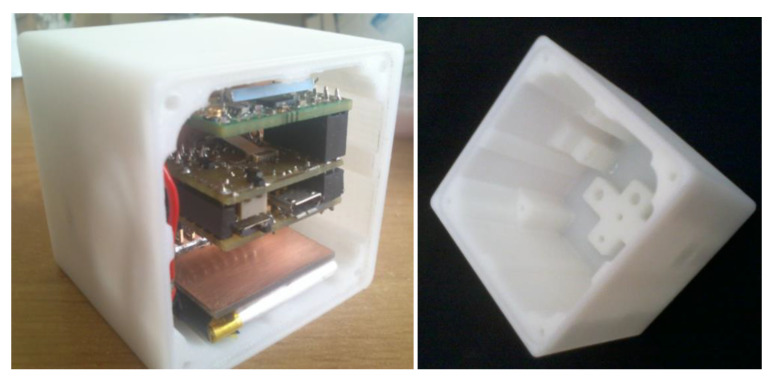
Tilt sensor with 3D-printed housing and wireless transmission.

**Figure 9 sensors-22-01504-f009:**
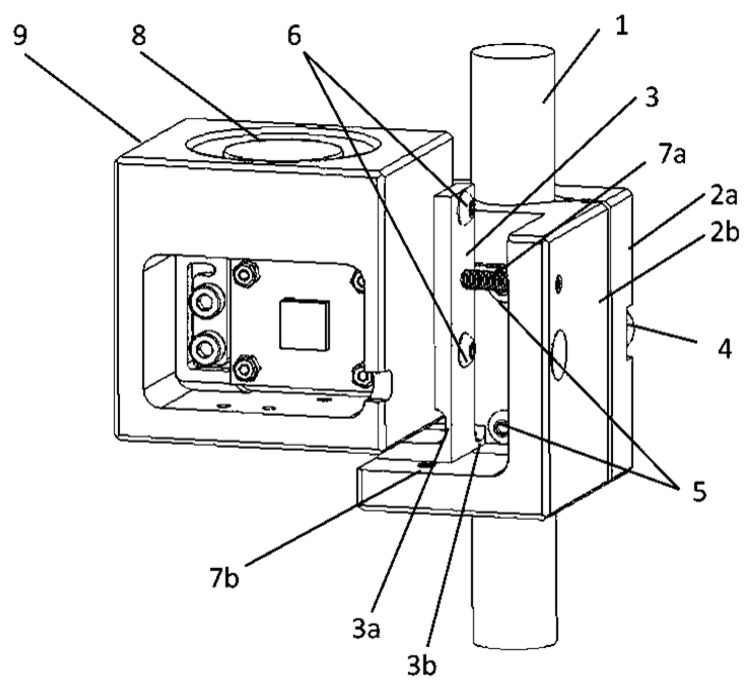
Device for mounting the tilt sensor on a lifting rope: 1—rope; 2a and 2b—two-part housing; 3—aligning fixture; 3a and 3b—constriction; 4 and 5 and 6—clamping screws; 7a and 7b—aligning screws; 8—spirit level; 9—housing of the tilt sensor.

**Figure 10 sensors-22-01504-f010:**
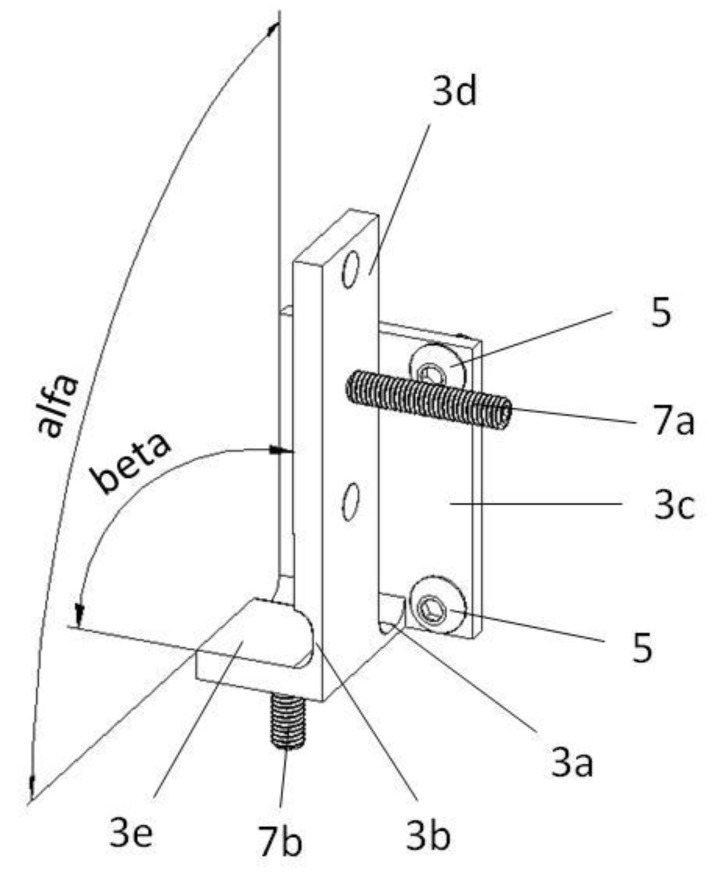
Compliant fixture: 3a and 3b—constriction; 3c and 3d—mounting wall; 3e—working wall; 5—clamping screws; 7a and 7b—aligning screws.

**Table 1 sensors-22-01504-t001:** Averaged values of the output voltages just after start-up of the sensor.

Position	*U_x_* (V)	*U_y_* (V)	*U_z_* (V)
#1	1.0593	1.4923	1.5159
#2	1.8938	1.4846	1.5004
#3	1.4865	1.4654	1.9270
#4	1.4673	1.5118	1.0934
#5	1.4660	1.0741	1.4859
#6	1.4890	1.9024	1.5327

**Table 2 sensors-22-01504-t002:** Averaged values of the output voltages one hour after the start-up.

Position	*U_x_* (V)	*U_y_* (V)	*U_z_* (V)
#1	1.0592	1.4919	1.5169
#2	1.8942	1.4845	1.5017
#3	1.4870	1.4646	1.9267
#4	1.4673	1.5103	1.0931
#5	1.4664	1.0743	1.4854
#6	1.4892	1.9026	1.5303

**Table 3 sensors-22-01504-t003:** Averaged values of the output voltages two hours after the start-up.

Position	*Ux* (V)	*Uy* (V)	*Uz* (V)
#1	1.0594	1.4922	1.5160
#2	1.8938	1.4848	1.5005
#3	1.4867	1.4652	1.9268
#4	1.4669	1.5100	1.0933
#5	1.4646	1.0743	1.4856
#6	1.4810	1.9025	1.5332

**Table 4 sensors-22-01504-t004:** Averaged values of the output voltages four hours after the start-up.

Position	*U_x_* (V)	*U_y_* (V)	*U_z_* (V)
#1	1.0594	1.4919	1.5157
#2	1.8938	1.4845	1.4999
#3	1.4872	1.4652	1.9269
#4	1.4670	1.5099	1.0932
#5	1.4655	1.0742	1.4860
#6	1.4894	1.9023	1.5306

**Table 5 sensors-22-01504-t005:** Averaged values of the output voltages six hours after the start-up.

Position	*U_x_* (V)	*U_y_* (V)	*U_z_* (V)
#1	1.0596	1.4925	1.5157
#2	1.8941	1.4844	1.5002
#3	1.4870	1.465	1.9270
#4	1.4672	1.5112	1.0933
#5	1.4651	1.0746	1.4875
#6	1.4890	1.9025	1.5299

**Table 6 sensors-22-01504-t006:** Offset variations within six hours.

Time from the Start-Up (h)	*OF_x_* (V)	Δ*OF_x_* (%)	*OF_y_* (V)	Δ*OF_y_* (%)	*OF_z_* (V)	Δ*OF_z_* (%)
0	1.4765	0	1.4882	0	1.5102	0
1	1.4767	0.01%	1.4885	0.01%	1.5099	−0.02%
2	1.4766	0.01%	1.4884	0.01%	1.5101	−0.01%
4	1.4766	0.01%	1.4883	0.00%	1.5101	−0.01%
6	1.4768	0.02%	1.4886	0.02%	1.5101	0.00%
	Max.	0.02%	Max.	0.02%	Max.	0.02%

**Table 7 sensors-22-01504-t007:** Scale factor variations within six hours.

Time from the Start-Up (h)	*SF_x_* (V/*g*)	Δ*SF_x_* (%)	*SF_y_* (V/*g*)	Δ*SF_y_* (%)	*SF_z_* (V/*g*)	Δ*SF_z_* (%)
0	0.4172	0	0.4141	0	0.4168	0
1	0.4175	0.06%	0.4141	0.00%	0.4168	0.00%
2	0.4172	−0.01%	0.4141	0.00%	0.4167	−0.01%
4	0.4172	0.00%	0.4140	−0.01%	0.4169	0.02%
6	0.4173	0.01%	0.4140	−0.04%	0.4168	0.01%
	Max.	0.06%	Max.	0.04%	Max.	0.02%

**Table 8 sensors-22-01504-t008:** Variation of the offset *OF* and the scale factor *SF* within 48 h.

Time from the Start-Up (h)	Δ*OF_x_* (%)	Δ*SF_x_* (%)	Δ*OF_y_* (%)	Δ*SF_y_* (%)	Δ*OF_z_* (%)	Δ*SF_z_* (%)
24	−0.01%	−0.01%	−0.03%	−0.03%	0.04%	0.00%
48	0.02%	−0.04%	0.00%	−0.03%	−0.20%	−0.02%
Max.	0.02%	0.04%	0.03%	0.03%	0.20%	0.02%

**Table 9 sensors-22-01504-t009:** Misalignments between the sensitive axes and the packaging (frame) [31,37,38].

Misalignment	ADIS 16488A (Deg)	ADIS16448 (Deg)	ADIS16240 (Deg)	ADIS16362 (Deg)	ADXL 327 (Deg)	Min. (Deg)	Max. (Deg)
Sensor type	10 DOF IMU	10 DOF IMU	10 DOF IMU	6 DOF IMU	accelerometer		
Axis to axis	±0.035	±0.05	±0.1	±0.2	±0.1	±0.035	±0.2
Axis to frame	±1	±0.5	±1	±0.5	±1	±0.5	±1

## Data Availability

Data available upon request.

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
