# Peer review of "Tilt Sensor with Recalibration Feature Based on MEMS Accelerometer"

_sensors, 2022, doi:10.3390/s22041504_

Round 1

Reviewer 1 Report

In this work, an aligning unit for aligning orientation of the accelerometer sensitive axes, and a solid cubic housing for re-calibration, are used to improve the disadvantages of MEMS accelerometer based tilt sensor. Due to the easy and quick re-calibration procedure, the thermal and long-term drifts, effects of aging can be effectively decreased while keeping the sensitive axes aligned with respect to the walls of the housing used as external mechanical datum. The concept of this work is novel, and has a very positive significance to the application of this kind of tilt sensor. No obvious problems have been found. I suggest to accept this paper.

Author Response

"In this work, an aligning unit for aligning orientation of the accelerometer sensitive axes, and a solid cubic housing for re-calibration, are used to improve the disadvantages of MEMS accelerometer based tilt sensor. Due to the easy and quick re-calibration procedure, the thermal and long-term drifts, effects of aging can be effectively decreased while keeping the sensitive axes aligned with respect to the walls of the housing used as external mechanical datum. The concept of this work is novel, and has a very positive significance to the application of this kind of tilt sensor. No obvious problems have been found. I suggest to accept this paper. "

*********

Thank you very much for the favorable opinion about the manuscript. According to suggestions of other reviewers, we decreased the rate of self-citations, added detailed information about the used measurement equipment, included detailed error analyses, revised Abstract, Introduction and Summary.

Reviewer 2 Report

I just suggest to give a re-reading and insert a comparison chart with other similar methodologies or sensors. I also find it appropriate to insert a more detailed paragraph on the signal conditioning and acquisition part, as these elements can also affect the system.

Author Response

"I just suggest to give a re-reading and insert a comparison chart with other similar methodologies or sensors. I also find it appropriate to insert a more detailed paragraph on the signal conditioning and acquisition part, as these elements can also affect the system."

*********

Thank you very much for the apt remarks. Please note that we added Section 3.1 and expanded Section 4 in the revised manuscript, providing many details about the equipment used in the experimental study. In Summary we included the proposed comparison table.

Reviewer 3 Report

Very interesting article. Authors describe in details recalibration method of the MEMS accelerometer. They present two housing dedicated to this aim.

One remark. The self-citation ratio is to high. It should not exceed 10% of the references.

Author Response

"Very interesting article. Authors describe in details recalibration method of the MEMS accelerometer. They present two housing dedicated to this aim.

One remark. The self-citation ratio is to high. It should not exceed 10% of the references. "

*********

Thank you very much for the favourable opinion about the manuscript. According to the suggestion, we have removed 2 self-publications, leaving 4 self-publications (not counting the patents); 5 new references were added. According to suggestions of other reviewers, we added detailed information about the used measurement equipment, included detailed error analyses, revised Abstract, Introduction and Summary.

Reviewer 4 Report

This research is missing the hypothesis part.  It is not clear what the authors' objectives are. A 3D housing with three orthogonal set-screws is not novel, nevertheless, it still poses some challenges which can be studied. For example, one can include a study about how a change in one set-screw setting influences the other axes - the 'cross coupling' effect. This study would also imply how to 'zero'  a 3D sensor which can be non-trivial. The merit of reporting  on a 3D printed enclosure at the end is not clear. 3D printed enclosures are not suitable for a precision sensor housing due to their production, thermal and long-term instability. 

Otherwise, the paper is well written and reads well. The critique above is to strengthen the work outcome of this research. 

Author Response

This research is missing the hypothesis part.  It is not clear what the authors' objectives are. A 3D housing with three orthogonal set-screws is not novel, nevertheless, it still poses some challenges which can be studied. For example, one can include a study about how a change in one set-screw setting influences the other axes - the 'cross coupling' effect. This study would also imply how to 'zero'  a 3D sensor which can be non-trivial. The merit of reporting  on a 3D printed enclosure at the end is not clear. 3D printed enclosures are not suitable for a precision sensor housing due to their production, thermal and long-term instability. 

Otherwise, the paper is well written and reads well. The critique above is to strengthen the work outcome of this research. 

*********

Thank you very much for the favourable opinion about the manuscript in the review. According to the suggestion, we have introduced a kind of a hypothesis in lines 69-76. With regard to the 'cross coupling' effect , we added some comments in lines 199-204. We expanded Section 7 about the 3-D prints in lines: 462-465 and 488-498. According to suggestions of other reviewers, we added detailed information about the used measurement equipment, included detailed error analyses, revised Abstract, Introduction and Summary.

Reviewer 5 Report

A brief summary

Sergiusz Łuczak, Maciej Zams, Bogdan Dąbrowski, and Zbigniew Kusznierewicz present a two-axial tilt sensor, whose mechanical structure makes it possible to perform a reliable and repeatable calibration of the accelerometer. The Paper is interesting, generally well written, and worth publishing. The work presents interesting results of research. However, it requires some improvement. First, the main purpose of the work was not clearly defined enough. The authors should clearly highlight the main aim of the work in the abstract first, and then in the introduction. The novelty of the described research should be emphasized in relation to the work of other authors and, in particular, to the authors' own works [4, 8, 10, 14, 29, 35, 36, 37]. The results of the experiments have not been described in sufficient detail. They are presented too broadly and imprecisely. A serious shortcoming at work is the lack uncertainty budget analysis. The authors completely omitted the analysis of accuracy, errors, and uncertainty of measurements. For this reason, the presented measurement results are not very reliable. Despite these shortcomings, the article may be of interest to many readers. However, it requires some improvement. First, the methods used and the results of the experiments have not been described in sufficient detail. The research results obtained should be refined and better presented. Conclusions should be more detailed. It is worth extending the discussion of results to scientific aspects. Furthermore, there are also a few mistakes in editing the text. These shortcomings significantly reduce the value of the entire article. Therefore I believe that specific changes, additions, and revisions are necessary.

Broad comments

  1. Generally, Authors should once again go through MDPI recommendations included in the Instructions for Authors (https://www.mdpi.com/journal/sensors/instructions), sensors-template.dot file (https://www.mdpi.com/files/word-templates/ sensors-template.dot), Manuscript English Editing, Guidelines for Authors (https://www.mdpi.com/authors/english-editing) and make sure they comply with this recommendations. In particular, the authors should pay attention to the following MDPI recommendations:

- In the abstract, the authors should clearly highlight the purpose of the work, and summarize the article's main findings,

- The abstract should be an objective representation of the article: it must not contain results which are not presented and substantiated in the main text,

- The introduction should define the purpose of the work and its significance, including specific hypotheses being tested. The main aim of the work should be mentioned and the main conclusions highlighted,

- Materials and methods should be described with sufficient detail to allow others to replicate and build on published results,

- A concise and precise description of the experimental results should be provided, their interpretation as well as the experimental conclusions that can be drawn.

- All words in headings should be capitalized,

- A figure caption on a single line should be centered,

- Equations should be punctuated as regular text,

- All symbols representing physical quantities and variables should be italicized,

  1. What is the purpose of this work? This is not clearly stated. First, the authors should clearly highlight the purpose of the work in the abstract. Then, the main aim of the work should be briefly mentioned in the introduction.
  2. The abstract should be an objective representation of the article. It should follow the style of structured abstracts: place the question addressed in a broad context and highlight the purpose of the study, describe briefly the main methods, summarize the article's main findings, and indicate the main conclusions or interpretations. The authors wrote: “it is possible to align the orientation of the accelerometer sensitive axes with respect to the housing with accuracy of at least 0.1 °”, but this has not been proven in the article. The abstract must not contain results which are not presented and substantiated in the main text. Therefore, the abstract should be redrafted, corrected, and improved.
  3. The introduction should briefly define the purpose of the work and its significance, including specific hypotheses being tested. The authors should briefly mention the main aim of the work and highlight the main conclusions. Therefore, the introduction needs to be corrected and supplemented.
  4. The results of the research are quite interesting and it seems that the proposed method can be widely used in the future. However, there is no clearly expressed scientific aspect in the work. In its current form, the work looks like a very extended test report. The article should specify the scientific problem that has been solved.
  5. A serious shortcoming at work is the lack uncertainty budget analysis. The tables show the measurement results with very high accuracy. These results are unreliable and raise doubts. What measuring equipment was used? What variations in the results occurred in the series of measurements? Type A and type B uncertainties should be reported. Measurement results should be presented with a resolution appropriate to the uncertainty. Has it been done so? This is not known. The authors completely missed the measurement uncertainty problem.
  6. Thermal and long-term drift studies are also questionable. It should be noted that the sensitivity and zero g bias level are essentially ratiometric to supply voltage. The authors ignored the problem of the stability of the voltage supplying the sensor. Perhaps the measurement results show the instability of the supply voltage? This is not known. The authors are silent about this.
  7. The authors present many measurement results of the sensor output voltages, their temperature, and long-term drifts. But the article is about the tilt sensor. So, interesting are the errors in the tilt measurement. It should be shown how voltage drifts are translated into tilt angle drifts?
  8. What 3D printing technology was used? What construction material was used? What is the accuracy of 3D printing of the technology used?
  9. In the summary, authors present tilt sensor parameters: measurement range: 360 degrees arc about two axes, measurement error approx. 0.2 degrees arc, and misalignments of the sensitive axes with respect to the housing of approx. 0.1 degree arc. How were these parameters tested and measured? This article does not cover this issue. The summary may only contain the results presented and justified in the main text.
  10. Punctuation in sentences containing equations should be improved. According to the "sensors-template.dot", equations should be punctuate as regular text. All equations and other mathematical expressions, both in running text and when displayed on separate lines, should be accompanied with appropriate punctuation, according to their function in the sentence. For example, a comma should be written after equation (1) immediately. Likewise in other places.

Specific comments

Lines 11-19. Abstract.

What is the main aim of this work? This is not clearly stated. In the abstract, the main purpose of the work was not clearly defined enough. The authors should clearly highlight the purpose of the work, and summarize the article's main findings. The authors wrote: “By means of the aligning unit it is possible to align orientation of the accelerometer sensitive axes with respect to the housing with accuracy of at least 0.1 degree” This has not been investigated at work. There is no justification for this conclusion.

Lines 22-81. Introduction.

The introduction has been written very well and provides a good background. But, in the introduction, the authors should explain the main aim of the work and finally highlight the main conclusions. The authors wrote: “the high accuracy of the sensor readings is achieved by using a special housing that allows a periodic quick and very simple calibration of the accelerometer to be performed at the user's request.” What does “high accuracy” mean? Is this really true? The authors did not conduct any research in this direction. On what basis do they make this conclusion? So, the introduction needs to be corrected and supplemented.

Line 82, 94, 139, and others. All words in headings should be capitalized.

Lines 233, 242 and others. Equations should be punctuated as regular text. A comma should be written after equation (1) immediately. Likewise in other places.

Line 233, equation (1). All symbols representing physical quantities and variables should be italicized. Likewise in other places.

Lines 184, 223 and others. A figure caption on a single line should be centered.

Lines 264-344, Thermal and long-term drifts. What accuracy of measurements was obtained? What are the measurement errors? What are the uncertainties? What is the nonlinearity? This should be calculated and stated in the article. Then it will be the scientific research. Materials and methods should be described with sufficient detail to allow others to replicate and build on published results. The authors should provide the types of apparatus used, manufacturers, specification of the main parameters (measuring ranges, accuracy). Without this data, the article is worthless to the reader.

Line 428-433. These conclusions are questionable. These conclusions are not supported by any research.

Author Response

Please, see the attached PDF file. Regards

Round 2

Reviewer 5 Report

Sergiusz Łuczak, Maciej Zams, Bogdan Dąbrowski, and Zbigniew Kusznierewicz present a revised and supplemented version of their previous article. The authors took into account most of the suggested comments and revised their manuscript accordingly. The authors' answer is satisfactory to me. There is only one small remark regarding the text. Authors should correct this error in the final manuscript (see comment below). I propose to accept this paper after this correction.

Lines 198, 249, 267 469. A figure caption on a single line should be centered.